# NEURAL MAP: STRUCTURED MEMORY FOR DEEP REINFORCEMENT LEARNING

**Emilio Parisotto & Ruslan Salakhutdinov**
Department of Machine Learning
Carnegie Mellon University
Pittsburgh, PA 15213, USA
`{eparisot,rsalakhu}@cs.cmu.edu`

## ABSTRACT

A critical component to enabling intelligent reasoning in partially observable environments is memory. Despite this importance, Deep Reinforcement Learning (DRL) agents have so far used relatively simple memory architectures, with the main methods to overcome partial observability being either a temporal convolution over the past $k$ frames or an LSTM layer. More recent work (Oh et al., 2016) has went beyond these architectures by using memory networks which can allow more sophisticated addressing schemes over the past $k$ frames. But even these architectures are unsatisfactory due to the reason that they are limited to only remembering information from the last $k$ frames. In this paper, we develop a memory system with an adaptable write operator that is customized to the sorts of 3D environments that DRL agents typically interact with. This architecture, called the Neural Map, uses a spatially structured 2D memory image to learn to store arbitrary information about the environment over long time lags. We demonstrate empirically that the Neural Map surpasses previous DRL memories on a set of challenging 2D and 3D maze environments and show that it is capable of generalizing to environments that were not seen during training.

## 1 INTRODUCTION

Memory is a crucial aspect of an intelligent agent's ability to plan and reason in partially observable environments. Without memory, agents must act reflexively according only to their immediate percepts and cannot execute plans that occur over an extended time interval. Recently, Deep Reinforcement Learning agents have been capable of solving many challenging tasks such as Atari Arcade Games (Mnih et al., 2015), robot control (Levine et al., 2016) and 3D games such as Doom (Lample & Chaplot, 2016), but successful behaviours in these tasks have often only been based on a relatively short-term temporal context or even just a single frame. On the other hand, many tasks require long-term planning, such as a robot gathering objects or an agent searching a level to find a key in a role-playing game.

Neural networks that utilized external memories have recently had an explosion in variety, which can be distinguished along two main axes: memories with write operators and those without. Writeless external memory systems, often referred to as "Memory Networks" (Sukhbaatar et al., 2015; Oh et al., 2016), typically fix which memories are stored. For example, at each time step, the memory network would store the past M states seen in an environment. What is learnt by the network is therefore how to access or read from this fixed memory pool, rather than what contents to store within it.

The memory network approach has been successful in language modeling, question answering (Sukhbaatar et al., 2015) and was shown to be a sucessful memory for deep reinforcement learning agents in complex 3D environments (Oh et al., 2016). By side-stepping the difficulty involved in learning what information is salient enough to store in memory, the memory network introduces two main disadvantages. The first disadvantage is that a potentially significant amount of redundant information could be stored. The second disadvantage is that a domain expert must choose what to store in the memory, e.g. for the DRL agent, the expert must set M to a value that is larger than the time horizon of the currently considered task.

On the other hand, external neural memories having write operations are potentially far more efficient, since they can learn to store salient information for unbounded time steps and ignore any other useless information, without explicitly needing any a priori knowledge on what to store. One prominent research direction within write-based architectures has been neural memories based on the types of memory structures that are found in computers, such as tapes, RAM, and GPUs. In contrast to typical recurrent neural networks, these neural computer emulators have far more structured memories which follow many of the same design paradigms that digital computers have traditionally utilized. One such model, the Differentiable Neural Computer (DNC) (Graves et al., 2016) and its predecessor the Neural Turing Machine (NTM) (Graves et al., 2014), structure the architecture to explicitly separate memory from computation. The DNC has a recurrent neural controller that can access an external memory resource by executing differentiable read and write operations. This allows the DNC to act and memorize in a structured manner resembling a computer processor, where read and write operations are sequential and data is store distinctly from computation. The DNC has been used sucessfully to solve complicated algorithmic tasks, such as finding shortest paths in a graph or querying a database for entity relations.

Building off these previous external memories, we introduce a new architecture called the Neural Map, a structured memory designed specifically for reinforcement learning agents in 3D environments. The Neural Map architecture overcomes some of the shortcomings of the previously mentioned neural memories. First, it uses an adaptable write operation and so its size and computational cost does not grow with the time horizon of the environment as it does with memory networks. Second, we impose a particular inductive bias on the write operation so that it is 1) well suited to 3D environments where navigation is a core component of sucessful behaviours, and 2) uses a sparse write operation that prevents frequent overwriting of memory locations that can occur with NTMs and DNCs. To accomplish this, we structure a DNC-style external memory in the form of a 2-dimensional map, where each position in the map is a distinct memory.

To demonstrate the effectiveness of the neural map, we run it on a variety of 2D partially-observable maze-based environments and test it against LSTM and memory network policies. Finally, to establish its scalability, we run a Neural Map agent on a set of challenging 3D maze environments based on the video game Doom.

## 2 BACKGROUND

A Markov Decision Process (MDP) is defined as a tuple $(\mathcal{S}, \mathcal{A}, \mathcal{T}, \gamma, \mathcal{R})$ where $\mathcal{S}$ is a finite set of states, $\mathcal{A}$ is a finite set of actions, $\mathcal{T}(s'|s, a)$ is the transition probability of arriving in state $s'$ when executing action $a$ in initial state $s$, $\gamma$ is a discount factor, and $\mathcal{R}(s, a, s')$ is the reward function of executing action $a$ in state $s$ and ending up at state $s'$. We define a policy $\pi(\cdot|s)$ as a mapping from a state $s$ to a distribution over actions, where $\pi(a_i|s)$ denotes the probability of action $a_i$ given that we are in state $s$. The value of a policy $V^\pi(s)$ is the expected discounted cumulative reward when starting from state $s$ and sampling actions according to $\pi$, i.e.: $V^\pi(s) = \mathbb{E}_\pi \left[ \sum_{t=0}^\infty \gamma^t R_t | s_0 = s \right]$.

An optimal value function, denoted $V^*(s)$, is the maximum value we can get from state $s$ according to any policy, i.e. $V^*(s) = \max_\pi V^\pi(s)$. An optimal policy $\pi^*$ is defined as a policy which achieves optimal value at each state, i.e. $V^{\pi^*}(s) = V^*(s)$. An optimal policy is guaranteed to exist (Sutton & Barto, 1998). The REINFORCE algorithm (Williams, 1992) iteratively updates a given policy $\pi$ in the direction of the optimal policy. This update direction is defined by $\nabla_\pi \log \pi(a_t|s_t)G_t$ with $G_t = \sum_{k=0}^\infty \gamma^k R_{t+k}$ being the future cumulated reward for a particular episode rollout. The variance of this update is typically high but can be reduced by using a "baseline" $b_t(s_t)$, which is a function of the current state. Therefore the baseline-augmented update equation is $\nabla_\pi \log \pi(a_t|s_t)(G_t - b_t(s_t))$. The typically used baseline is the value function, $b_t(s_t) = V^\pi(s_t)$. This combination of REINFORCE with value function baseline is commonly termed the "Actor-Critic" algorithm.

In this paper, we utilize Advantage Actor-Critic (A2C) (Mnih et al., 2016) with Generalized Advantage Estimation (Schulman et al., 2015), which can be seen as a specialization of the actor-critic framework when using deep networks to parameterize the policy and value function. The policy is a function of the state, parameterized as a deep neural network: $\pi(a|s) = f_\theta(s, a)$, where f is a deep neural network with parameter vector $\theta$.

## 3   NEURAL MAP

In this section, we will describe the details of the neural map. We assume we want our agent to act within some 2- or 3-dimensional environment. The neural map is the agent's internal memory storage that can be read from and written to during interaction with its environment, but where the write operator is selectively limited to affect only the part of the neural map that represents the area where the agent is currently located. For this paper, we assume for simplicity that we are dealing with a 2-dimensional map. This can easily be extended to 3-dimensional or even higher-dimensional maps (i.e. a 4D map with a 3D sub-map for each cardinal direction the agent can face).

Let the agent's position be $(x, y)$ with $x \in \mathbb{R}$ and $y \in \mathbb{R}$ and let the neural map $M$ be a $C \times H \times W$ feature block, where $C$ is the feature dimension, $H$ is the vertical extent of the map and $W$ is the horizontal extent. Assume there exists some coordinate normalization function $\psi(x, y)$ such that every unique $(x, y)$ can be mapped into $(x', y')$, where $x' \in \{0, \dots, W-1\}$ and $y' \in \{0, \dots, H-1\}$. For ease of notation, suppose in the sequel that all coordinates have been normalized by $\psi$ into neural map space.

Let $s_t$ be the current state embedding, $M_t$ be the current neural map, and $(x_t, y_t)$ be the current position of the agent. The Neural Map is defined by the following set of equations:

$$r_t = read(M_t), \quad c_t = context(M_t, s_t, r_t),$$
$$w_{t+1}^{(x_t,y_t)} = write(s_t, r_t, c_t, M_t^{(x_t,y_t)}), \quad M_{t+1} = update(M_t, w_{t+1}^{(x_t,y_t)}),$$
$$o_t = [r_t, c_t, w_{t+1}^{(x_t,y_t)}], \quad \pi_t(a|s) = \text{Softmax}(f(o_t)), \tag{1}$$

where $w_t^{(x_t,y_t)}$ represents the feature at position $(x_t, y_t)$ at time $t$, $[x_1, \dots, x_k]$ represents a concatenation operation, and $o_t$ is the output of the neural map at time $t$ which is then processed by another deep network $f$ to get the policy outputs $\pi_t(a|s)$. We will now separately describe each of the above operations in more detail:

**Global Read Operation:** The $read$ operation passes the current neural map $M_t$ through a deep convolutional network and produces a $C$-dimensional feature vector $r_t$. The global read vector $r_t$ summarizes information about the entire map.

**Context Read Operation:** The $context$ operation performs context-based addressing to check whether certain features are stored in the map. It takes as input the current state embedding $s_t$ and the current global read vector $r_t$ and first produces a query vector $q_t$. The inner product of the query vector and each feature $M_t^{(x,y)}$ in the neural map is then taken to get scores $a_t^{(x,y)}$ at all positions $(x, y)$. The scores are then normalized to get a probability distribution $\alpha_t^{(x,y)}$ over every position in the map, also known as "soft attention" (Bahdanau et al., 2015). This probability distribution is used to compute a weighted average $c_t$ over all features $M_t^{(x,y)}$. To summarize:

$$q_t = W[s_t, r_t], \quad a_t^{(x,y)} = q_t \cdot M_t^{(x,y)},$$
$$\alpha_t^{(x,y)} = \frac{e^{a_t^{(x,y)}}}{\sum_{(w,z)} e^{a_t^{(w,z)}}}, \quad c_t = \sum_{(x,y)} \alpha_t^{(x,y)} M_t^{(x,y)}, \tag{2}$$

where $W$ is a weight matrix. The context read operation allows the neural map to operate as an associative memory: the agent provides some possibly incomplete memory (the query vector $q_t$) and the operation will return the completed memory that most closely matches $q_t$. So, for example, the agent can query whether it has seen something similar to a particular landmark that is currently within its view.

**Local Write Operation:** Given the agent's current position $(x_t, y_t)$ at time $t$, the $write$ operation takes as input the current state embedding $s_t$, the global read output $r_t$, the context read vector $c_t$ and the current feature at position $(x_t, y_t)$ in the neural map $M_t^{(x_t,y_t)}$ and produces, using a deep neural network $f_w$, a new C-dimensional vector $w_{t+1}^{(x_t,y_t)}$. This vector functions as the new local write candidate vector at the current position $(x_t, y_t)$: $w_{t+1}^{(x_t,y_t)} = f_w([s_t, r_t, c_t, M_t^{(x_t,y_t)}])$

**GRU-based Local Write Operation** As previously defined, the write operation simply replaces the vector at the agent's current position with a new feature produced by a deep network. Instead of this

hard rewrite of the current position's feature vector, we can use a gated write operation based on the recurrent update equations of the Gated Recurrent Unit (GRU) (Chung et al., 2014). Gated write operations have a long history in unstructured recurrent networks and they have shown a superior ability to maintain information over long time lags versus ungated networks. The GRU-based write operation is defined as:

$$r_{t+1}^{(x_t,y_t)} = \sigma(W_r[s_t, r_t, c_t, M_t^{(x_t,y_t)}])$$
$$\hat{w}_{t+1}^{(x_t,y_t)} = \tanh(W_{\hat{h}}[s_t, r_t, c_t] + U_{\hat{h}}(r_{t+1}^{(x_t,y_t)} \odot M_t^{(x_t,y_t)}))$$
$$z_{t+1}^{(x_t,y_t)} = \sigma(W_z[s_t, r_t, c_t, M_t^{(x_t,y_t)}])$$
$$w_{t+1}^{(x_t,y_t)} = (1 - z_{t+1}^{(x_t,y_t)}) \odot M_t^{(x_t,y_t)} + z_{t+1}^{(x_t,y_t)} \odot \hat{w}_{t+1}^{(x_t,y_t)},$$

where $x \odot y$ is the Hadamard product between vectors $x$ and $y$, $\sigma(\cdot)$ is the sigmoid activation function and $W_*$ and $U_*$ are weight matrices. Using GRU terminology, $r_{t+1}^{(x_t,y_t)}$ is the reset gate, $\hat{w}_{t+1}^{(x_t,y_t)}$ is the candidate activation and $z_{t+1}^{(x_t,y_t)}$ is the update gate. By making use of the reset and update gates, the GRU-based update can modulate how much the new write vector should differ from the currently stored feature.

**Map Update Operation:** The *update* operation creates the neural map for the next time step. The new neural map $M_{t+1}$ is equal to the old neural map $M_t$, except at the current agent position $(x_t, y_t)$, where the current write candidate vector $w_{t+1}^{(x_t,y_t)}$ is stored:

$$M_{t+1}^{(a,b)} = \begin{cases} w_{t+1}^{(x_t,y_t)}, & \text{for } (a,b) = (x_t, y_t) \\ M_t^{(a,b)}, & \text{for } (a,b) \neq (x_t, y_t) \end{cases} \tag{3}$$

# 4 EGO-CENTRIC NEURAL MAP

A major disadvantage of the neural map as previously described is that it requires some oracle to provide the current $(x, y)$ position of the agent. This is a difficult problem in and of itself, and, despite being well studied, it is far from solved. The alternative to using absolute positions within the map is to use relative positions. That is, whenever the agent moves between time steps with some velocity $(u, v)$, the map is counter-transformed by $(-u, -v)$, i.e. each feature in the map is shifted in the $H$ and $W$ dimensions. This will mean that the map will be ego-centric, i.e. the agent's position will stay stationary in the center of the neural map while the world as defined by the map moves around them. Therefore in this setup we only need some way of extracting the agent's velocity, which is typically a simpler task in real environments (for example, animals have inner ears and robots have accelerometers). Here we assume that there is some function $\xi(u', v')$ that discretizes the agent velocities $(u', v')$ so that they represent valid velocities within the neural map $(u, v)$. In the sequel, we assume that all velocities have been properly normalized by $\xi$ into neural map space.

Let $(pw, ph)$ be the center position of the neural map. The updated ego-centric neural map operations are shown below:

$$\overline{M}_t = CounterTransform(M_t, (u_t, v_t))$$
$$r_t = read(\overline{M}_t) \quad c_t = context(\overline{M}_t, s_t, r_t)$$
$$w_{t+1}^{(pw,ph)} = write(s_t, r_t, c_t, \overline{M}_t^{(pw,ph)}) \quad M_{t+1} = egoupdate(\overline{M}_t, w_{t+1}^{(pw,ph)})$$
$$o_t = [r_t, c_t, w_{t+1}^{(pw,ph)}] \quad \pi_t = \text{Softmax}(f(o_t))$$

Where $\overline{M}_t$ is the current neural map $M_t$ reverse transformed by the current velocity $(u_t, v_t)$ so that the agents map position remains in the center $(pw, ph)$.

**Counter Transform Operation:** The $CounterTransform$ operation transforms the current neural map $M_t$ by the inverse of the agent's current velocity $(u_t, v_t)$. Written formally:

$$\overline{M}_t^{(a,b)} = \begin{cases} M_t^{(a-u,b-v)}, & \text{for } (a-u) \in \{1,...,W\} \wedge (b-v) \in \{1,...,H\} \\ 0, & \text{else} \end{cases} \tag{4}$$

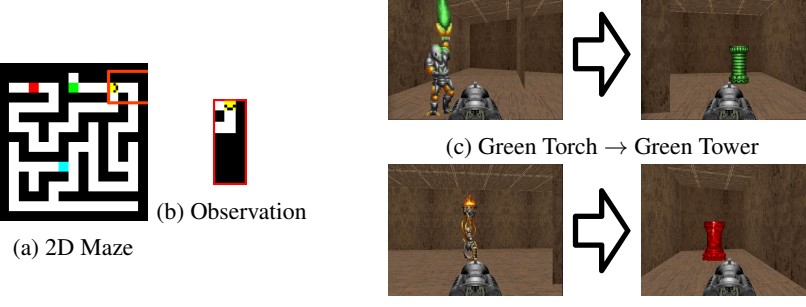

(c) Green Torch → Green Tower

(d) Red Torch → Red Tower

Figure 1: **Left:** Images showing the 2D maze environment. The left side (Fig. 1a) represents the fully observable maze while the right side (Fig. 1b) represents the agent observations. The agent is represented by the yellow pixel with its orientation indicated by the black arrow within the yellow block. The starting position is always the topmost position of the maze. The red bounding box represents the area of the maze that is subsampled for the agent observation. In "Goal-Search", the goal of the agent is to find a certain color block (either red or teal), where the correct color is provided by an indicator (either green or blue). This indicator has a fixed position near the start position of the agent. **Right:** State observations from the "Indicator" Doom maze environment. The agent starts in the middle of a maze looking in the direction of a torch indicator. The torch can be either green (top-left image) or red (bottom-left image) and indicates which of the goals to search for. The goals are two towers which are randomly located within the maze and match the indicator color. The episode ends whenever the agent touches a tower, whereupon it receives a positive reward if it reached the correct tower, while a negative reward otherwise.

While here we only deal with reverse translation, it is possible to handle rotations as well if the agent can measure it's angular velocity.

**Map Egoupdate Operation:** The $egoupdate$ operation is functionally equivalent to the $update$ operation except only the center position $(pw, ph)$ is ever written to:

$$M_{t+1}^{(a,b)} = \begin{cases} w_{t+1}^{(pw,ph)}, & \text{for } (a,b) = (pw, ph) \\ \overline{M}_t^{(a,b)}, & \text{for } (a,b) \neq (pw, ph) \end{cases} \tag{5}$$

## 5   EXPERIMENTS

To demonstrate the effectiveness of the Neural Map, we run it on 2D and 3D maze-based environments where memory is crucial to optimal behaviour. We compare to previous memory-based DRL agents, namely a simple LSTM-based agent which consists of a single pre-output LSTM layer as well as MemNN (Oh et al., 2016) agents.

### 5.1   2D GOAL-SEARCH ENVIRONMENT

The "Goal-Search" environment is adapted from Oh et al. (2016). Here the agent starts in a fixed starting position within some randomly generated maze with two randomly positioned goal states. It then observes an indicator at a fixed position near the starting state (i.e. the green tile at the top of the maze in Fig. 1a). This indicator will tell the agent which of the two goals it needs to go to (blue indicator→teal goal, green indicator→red goal). If the agent goes to the correct goal, it gains a positive reward while if it goes to the incorrect goal it gains a negative reward. Therefore the agent needs to remember the indicator as it searches for the correct goal state. In depth details of the 2D environment are given in Appendix B. The mazes during training are generated using a random generator. A held-out set of 1000 random mazes is kept for testing. This test set therefore represents maze geometries that have never been seen during training, and measure the agent's ability to generalize to new environments.

The first baseline agent we evaluate is a recurrent network with 128 LSTM units. The other baseline is the MQN, which is a memory-network-based architecture that performs attention over the past K states it has seen (Oh et al., 2016). Both LSTM and MQN models receive a one-hot encoding of the agent's current location, previous velocity, and current orientation at each time step, in order

| | 2D Goal-Search | | | | | |
| | Train | | | Test | | |
| Agent | 7-11 | 13-15 | Total | 7-11 | 13-15 | Total |
|---|---|---|---|---|---|---|
| Random | 41.9% | 25.7% | 38.1% | 46.0% | 29.6% | 38.8% |
| LSTM | 84.7% | 74.1% | 87.4% | 96.3% | 83.4% | 91.4% |
| MQN-32 | 80.2% | 64.4% | 83.3% | 95.9% | 74.6% | 87.4% |
| MQN-64 | 83.2% | 69.6% | 85.8% | 96.5% | 76.7% | 88.3% |
| Neural Map (15x15) | 92.4% | 80.5% | 89.2% | 93.5% | 87.9% | 91.7% |
| Neural Map + GRU (15x15) | 97.0% | 89.2% | 94.9% | 97.7% | 94.0% | 96.4% |
| Neural Map + GRU (8x8) | 94.9% | 90.7% | 95.6% | 98.0% | 95.8% | 97.3% |
| Neural Map + GRU + Pos (8x8) | 95.0% | 91.0% | 95.9% | 98.3% | 94.3% | 96.5% |
| Neural Map + GRU + Pos (6x6) | 90.9% | 83.2% | 91.8% | 97.1% | 90.5% | 94.0% |
| Ego Neural Map + GRU (15x15) | 94.6% | 91.1% | 95.4% | 97.7% | 92.1% | 95.5% |
| Ego Neural Map + GRU + Pos (15x15) | 74.6% | 63.9% | 78.6% | 87.8% | 73.2% | 82.7% |

Table 1: Results of several different agent architectures on the "Goal-Search" environment. The "train" columns represents the number of mazes solved (in %) when sampling from the same distribution as used during training. The "test" columns represents the number of mazes solved when run on a set of held-out maze samples which are guaranteed not to have been sampled during training.

to make the comparison to the fixed-frame Neural Map fair. We test these baselines against several Neural Map architectures, with each architecture having a different design choice.

The results are reported in Table 1. During testing, we extend the maximum episode length from 100 to 500 steps so that the agent is given more time to solve the maze. The brackets next to the model name represent the Neural Map dimensions of that particular model. From the results we can see that the Neural Map architectures solve the most mazes in both the training and test distributions compared to both LSTM and MQN baselines.

The results also demonstrate the effect of certain design decisions. One thing that can be observed is that using GRU updates adds several percentage points to the success rate ("Neural Map (15x15)" v.s. "Neural Map + GRU (15x15)"). We also tried downsampled Neural Maps, such that a pixel in the memory map represents several discrete locations in the environment. The Neural Map seems quite robust to this downsampling, with a downsampling of around 3 (6x6 v.s. 15x15) doing just a few percentage points worse, and still beating all baseline models. The 6x6 model has approximately the same number of memory cells as "MQN-32", but its performance is much better, showing the benefit of having learnable write operations. For the egocentric model, in order to cover the entire map we set the pixels to be 2x smaller in each direction, so each pixel is only a quarter of a pixel in the fixed-frame map. Even with this coarser representation, the egocentric model did similarly to the fixed frame one. We demonstrate an example of what the Neural Map learned to address using its context operator in Appendix E.

Finally, we tried adding the one-hot position encoding as a state input to the Neural Map, as is done for the baselines. We can see that there is a small improvement, but it is largely marginal, with the Neural Map doing a decent job of learning how to represent its own position without needing to be told explicitly. One interesting thing that we observed is that having the one-hot position encoding as an input to the egocentric map decreased performance, perhaps because it is difficult for the network to learn a mapping between fixed and egocentric frames.

Note that sometimes the percentage results are lower for the training distribution. This is mainly because the training set encompases almost all random mazes except the fixed 1000 of the test set, thus making it likely that the agent sees each training map only once.

Beyond train/test splits, the results are further separated by maze size. This information reveals that the memory networks are hardest hit by increasing maze size with sometimes a 20% drop in success on 13-15 v.s. 7-11. This is perhaps unsurprising given the inherent fixed time horizon of memory netwoks, and further reveals the benefit of using write-based memories.

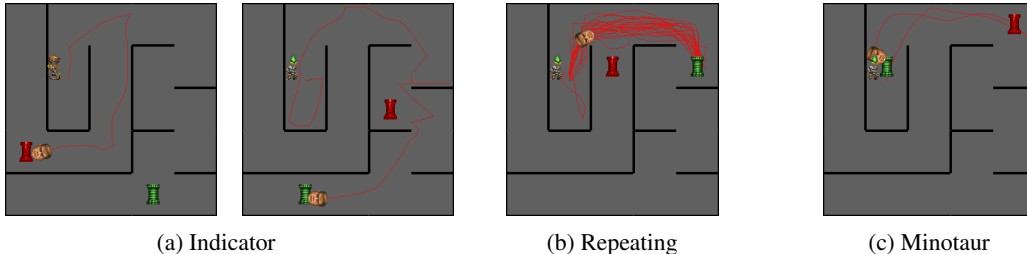

| (a) Indicator | (b) Repeating | (c) Minotaur |

Figure 2: Top-down views showing succesful episodes in each of the 3 Doom maze tasks. The red lines indicate the path traveled by the agent. **Indicator** is shown in Fig. 2a, where the agent receives positive reward when entering the corresponding tower that matches the torch color it saw at the start of the episode and a negative reward otherwise. The episode terminates once the agent has reached a tower. **Repeating**, shown in Fig. 2b, has the same underlying mechanics except (1) the episode persists for $T$ time steps regardless of towers entered and (2) the torch indicator is removed from the maze after the agent has reached a tower once. Therefore the agent needs to find the correct tower and then optimize its path to that tower. **Minotaur** shown in Fig. 2c requires the agent to reach the red goal and then return to the green goal that is at its starting position. Here the torch does not have any function. This fully-observable top-down view was not made available to the agent and is only used for visualization.

## 5.2 3D DOOM ENVIRONMENT DESCRIPTION

To demonstrate that our method can work in much more complicated 3D environments with longer time lags, we implemented three 3D maze environments using the ViZDoom (Kempka et al., 2016) API and a random maze generator. Examples of all three environments are given in Figure 2.

**Indicator Maze:** The first environment is a recreation of the 2D indicator maze task, where an indicator is positioned in view of the player's starting state which is either a torch of red or green color. The goals are corresponding red/green towers that are randomly positioned throughout the maze that the player must locate.

**Repeating Maze:** The second environment is a variant of this indicator maze but whenever the player enters a goal state, it is teleported back to the beginning of the maze without terminating the episode (i.e. it retains its memory of the current maze). It gains a positive reward if it reaches the correct goal and a negative reward if it reaches the incorrect goal. After the first goal is reached, the correct indicator color is no longer displayed within the maze and a red indicator is displayed afterwards instead (regardless if the correct goal is green). An episode ends after a predetermined number of steps which depends on the maze size. The goal is therefore to find a path to the correct goal, and then optimize that path so that it can reach it as many times as possible.

**Minotaur Maze:** The third environment has the agent start in a fixed starting position next to the green tower, while the red tower is randomly placed somewhere in the maze. The agent receives a small positive reward if it reaches the red tower, and a larger positive reward if after reaching the red tower it returns to the green tower. Therefore the agent must efficiently navigate to the red goal while accurately remember its entire path it so that it can backtrack to the start.

All three environments used a RGB+D image of size 100x60 as input. We generate maze geometries randomly at train time but make sure to exclude a test set of 10 mazes for each size [4, 5, 6, 7, 8] (50 total). For these environments, we tested out four architectures (see Appendix C for more details on both environments and architectures):

**Neural Map with Controller LSTM:** Standard Neural Map with fixed frame addressing and GRU updates. We combine the neural map design with an LSTM that aggregates past state, read and context vectors and produces the query vector for the next time step's context read operation. See Appendix A for the modified Neural Map equations.

**Ego Neural Map with Controller LSTM:** Same as previous but with ego-centric addressing. The other difference is that the Ego Neural Map does not receive any positional input unlike the other 3 models, only receiving frame-by-frame ego-motion (quantized to a coarse grid).

**LSTM:** Single pre-output 256-dimensional LSTM layer.

| Agent | | Indicator | | | | | Repeating | | | | | Minotaur | | | | |
|---|---|---|---|---|---|---|---|---|---|---|---|---|---|---|---|---|
| Maze Size | | 4 | 5 | 6 | 7 | 8 | 4 | 5 | 6 | 7 | 8 | 4 | 5 | 6 | 7 | 8 |
| LSTM | Acc | 95.7 | 87.5 | 81.1 | 71.4 | 60.3 | - | - | - | - | - | 90.0 | 71.5 | 48.0 | 34.2 | 29.4 |
| | Rew | - | - | - | - | - | 7.26 | 7.58 | 6.06 | 5.32 | 4.98 | 1.35 | 1.07 | 0.72 | 0.51 | 0.44 |
| FRMQN | Acc | 87.3 | 82.9 | 78.0 | 72.0 | 59.8 | - | - | - | - | - | 72.7 | 54.5 | 38.8 | 28.8 | 23.7 |
| | Rew | - | - | - | - | - | 1.45 | 1.65 | 1.51 | 1.37 | 1.09 | 1.09 | 0.82 | 0.58 | 0.43 | 0.36 |
| Controller NMap | Acc | **95.8** | 90.3 | 81.8 | 80.4 | 70.3 | - | - | - | - | - | **99.7** | **92.2** | **67.5** | 37.9 | 30.2 |
| | Rew | - | - | - | - | - | **17.4** | **17.1** | **12.0** | **11.4** | **12.3** | **1.50** | **1.38** | **1.01** | 0.57 | 0.45 |
| Controller Ego-NMap | Acc | 94.6 | **91.0** | **87.6** | **85.8** | **72.2** | - | - | - | - | - | 98.6 | 90.0 | 65.2 | **44.7** | **33.8** |
| | Rew | - | - | - | - | - | 12.8 | 14.1 | 11.0 | 10.4 | 9.72 | 1.48 | 1.35 | 0.98 | **0.67** | **0.51** |

Table 2: Doom results on mazes not observed during training for the three tasks: Indicator, Repeating and Minotaur. Acc stands for Accuracy and Rew for Reward. Accuracy for Indicator means % of correct goals reached, while for Minotaur it means % of episodes where the agent successfully reached the goal and then backtracked to the beginning. Reward for Repeating is number of times correct goal was visited within the allotted time steps (+1 for correct goal, -1 for incorrect goal). Reward for Minotaur is +0.5 for reaching the goal and then +1.0 for backtracking to start after reaching goal (max episode reward is +1.5). We tested on maze sizes between [4,8] with 10 test mazes for each size. For each of the 50 total test mazes we ran 100 episodes with random goal locations and averaged the result.

**FRMQN (Oh et al., 2016):** Memory network with LSTM feedback. This design uses an LSTM to make recurrent context queries to the memory network database. In addition, for the memory network baselines we did not set a fixed k but instead let it access any state from its entire episode. This means no information is lost to the memory network, it only needs to process its history.

The results are shown in Table 2. We can see that the Neural Map architectures work better than the baseline models, even though the memory network has access to its entire episode history at every time step. The ego-centric Neural Map beats the fixed frame map at Indicator, and gets similar performance on both Repeating and Minotaur environments, showing the ability of the Neural Map to function effectively even without global position information. It is possible that having a fixed frame makes path optimization easier, which would explain the larger rewards that the fixed-frame model got in the Repeating task. We also investigated whether the neural map is robust to localization noise, which would be the case in a real world setting where we do not have access to a localization oracle and must instead rely on an error-prone odometry or SLAM-type algorithm to do localization. These results are presented in Appendix D.

For the baselines, we can see that FRMQN has difficulty learning on Repeating, only reaching the goal on average once. This could be because the indicator is only shown before the first goal is reached and so afterwards it needs to remember increasingly longer time horizons. Furthermore, because the red indicator is always shown after the first goal is reached, it might be difficult for the model to learn to do retrieval since the original correct indicator must be indexed by time and not image similarity. The FRMQN also has difficulty on Minotaur, probably due to needing to remember and organize a lot of spatial information (i.e. what actions were taken along the path). For Indicator, the FRMQN does similarly to the LSTM. We can see that the spatial structure of the Neural Map aids in optimizing the path in Repeating, averaging 12 goal reaches even in the largest maze size.

## 6  RELATED WORK

Other than the straightforward architectures of combining an LSTM with Deep Reinforcement Learning (DRL) (Mnih et al., 2016; Hausknecht & Stone, 2015), there has also been work on using more advanced external memory systems with DRL agents to handle partial observability. Oh et al. (2016) used a memory network (MemNN) to solve maze-based environments similar to the ones presented in this paper. MemNN keeps the last $M$ states in memory and encodes them into (key, value) feature pairs. It then queries this memory using a soft attention mechanism similar to the context operation of the Neural Map, except in the Neural Map the key/value features were written by the agent and aren't just a stored representation of the last $M$ frames seen. Oh et al. (2016) tested a few variants of this basic model, including ones which combined both LSTM and memory-network style memories.

In contrast to memory networks, another research direction is to design recurrent architectures that mimic computer memory systems. These architectures explicitly separate computation and memory

in a way anagolous to a modern digital computer, in which some neural controller (akin to a CPU) interacts with an external memory (RAM). One recent model is similar to the Neural Map, called the Differentiable Neural Computer (DNC) (Graves et al., 2016), which combines a recurrent controller with an external memory system that allows several types of read/write access. In addition to defining an unconstrained write operator (in contrast to the neural map's write location being fixed), the DNC has a selective read operation that reads out the memory either by content or in the order that it was written. While the DNC is more specialized to solving algorithmic problems, the Neural Map can be seen as an extension of this Neural Computer framework to 3D environments, with a specific inductive bias on its write operator that allows sparse writes. Recently work has also been done toward sparsifying the read and write operations of the DNC (Rae et al., 2016). This work was not focused on 3D environments and did not make any use of task-specific biases like agent location, but instead used more general biases like "Least-Recently-Used" memory addresses to force sparsity. More recently, the DNC, in conjunction with a VIN planning network (Tamar et al., 2016), has been applied to the task of navigating partially-observable environments (Khan et al., 2018) although it still relied on supervised learning in order to train the complete system.

Gupta et al. (2017) designed a similar 2D map structured memory, with the aim to do robot navigation in 3D environments. These environments were based off image scans of real office buildings, and they were preprocessed into a grid-world by quantizing the possible positions and orientations the agent could assume. In contrast to our paper, which presents the Neural Map more as a general memory architecture for DRL agents, Gupta et al. (2017) focuses mainly on solving the task of robot navigation, with the internal map's representation mainly used to represent free space around the robot. More concretely, the task in these environments was to navigate to a goal state, with the goal position either stated semantically (find a chair) or stated in terms of the position relative to the robot's coordinate frame. Another key difference was that their formulation lacked a context addressing operation. Finally, their method used DAGGER (Ross et al., 2011), an imitation learning algorithm, to train their agent. Since Doom actions affect translational/rotational accelerations, training using imitation learning is more difficult since a search algorithm cannot be used directly as supervision. An interesting addition they made was the use of a multi-scale map representation and a Value Iteration network (Tamar et al., 2016) to do better path planning. Another related work, Neural SLAM (Zhang et al., 2017) extends spatial memories to settings where localization/odometry is not provided a priori, but instead has to be completed in tandem with the mapping of the environment. In order to accomplish that, a grid-based localization system was combined with a Neural Map-style memory in order to do differentiable SLAM-like combined localization and mapping.

## 7 CONCLUSION

In this paper we developed a neural memory architecture that organizes the spatial structure of its memory in the form of a 2D map, and allows sparse writes to this memory where the memory address of the write is in a correspondence to the agent's current position in the environment. We showed its ability to learn, using a reinforcement signal, how to behave within challenging 2D and 3D maze tasks that required storing information over long time steps. The results demonstrated that our architecture surpassed baseline memories used in previous work. Additionally, we showed the benefit of certain design decisions made in our architecture: using GRU updates instead of hard writes, demonstrating that the ego-centric viewpoint does not diminish performance and that the Neural Map is robust to downsampling its memory. Finally, to show that our method can scale up to more difficult 3D environments, we implemented several new maze environments in Doom. Using a hybrid Neural Map + LSTM model, we were able to solve most of the scenarios at a performance higher than previous DRL memory-based architectures. Furthermore, we demonstrated the ability of the Neural Map to be robust to a certain level of drift noise in its localization estimate.

**Acknowledgements**

This work was supported by Apple, DARPA award D17AP00001, the Google focused award. The authors would also like to thank NVidia NVAIL award for donating DGX-1 deep learning machine. This work used the Extreme Science and Engineering Discovery Environment (XSEDE), which is supported by National Science Foundation grant number OCI-1053575. Specifically, it used the Bridges system, which is supported by NSF award number ACI-1445606, at the Pittsburgh Supercomputing Center (PSC).

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

# A   CONTROLLER (EGO-)NEURAL MAP

Here we describe the modification to the Neural Map we utilized for the 3D maze tasks. We include an extra state $h$ that represents the hidden and cell state of an LSTM. The Neural Map equations are therefore:

$$r_t = read(M_t), \tag{6}$$

$$h_t = LSTM(s_t, r_t, c_{t-1}, h_{t-1}), \tag{7}$$

$$c_t = context(M_t, h_t), \tag{8}$$

$$w_{t+1}^{(x_t, y_t)} = write(s_t, r_t, c_t, M_t^{(x_t, y_t)}), \tag{9}$$

$$M_{t+1} = update(M_t, w_{t+1}^{(x_t, y_t)}), \tag{10}$$

$$o_t = [r_t, c_t, w_{t+1}^{(x_t, y_t)}], \tag{11}$$

$$\pi_t(a|s) = \text{Softmax}(f(o_t)), \tag{12}$$

# B   2D ENVIRONMENT DETAILS

The state input for the 2D environment is a $5 \times 15 \times 3$ subsample of the complete maze so that the agent is able to see 15 pixel forward and 3 pixels on the side (center pixel + one pixel on each side of the agent) which is depicted in Fig.1b. This view is obscured so the agent is prevented from seeing the identity of anything behind walls. The 5 binary channels in the observation represent object identities: channel 1 represents presence of walls, 2 represents the green indicator, 3 the blue indicator, 4 the red goal, and 5 the teal goal. The LSTM and MemNN networks were given auxiliary information such as the one-hot encoding of the current agent's true position in the maze. Neural Map variants were not given position information in an auxiliary state unless specified by the "+ Pos" modifier in Table 1. Actions in the environment included moving forward and turning left/right 90 degrees.

For optimization, all architectures used the RMSprop optimization algorithm with gradients thresholded to norm 20 for LSTM, 100 for Neural Map variants, and no thresholding for memory networks. We used an auxiliary weighted entropy loss on the Synchronous Actor-Critic with weight 0.01. The learning rates for LSTM models was 0.0025, 0.005 for Neural Map variants, and 0.001 for memory networks. We obtained the hyperparameters during a limited hyperparameter sweep on a simpler version of the environment. We used A2C with number of time steps equal to 5. We trained for 10 million updates.

The mazes were generated using an algorithm based on Depth-First Search to form a fully-connected maze. Afterwards each wall was deleted with probability $p$. $p$ was sampled uniformly from between $[0, 0.75]$ at maze generation time. Each episode, a random maze width was sampled uniformly from between $[7, 15]$.

**(Ego-)Neural Map Agent Details:** For the global read operation, we used a convolutional network with 3 layers of 8 channels and kernel size 3. The strides were set to 1 on the first layer, and 2 to the second and third layers. Padding was 1 on the first layer and 0 on other layers. The 3 convolutional layers were then followed by a fully-connected layer of dimension 256 and then another fully-connected layer of size 32. All activations were relu except the 32 dimension layer, which was set to tanh. Positions were normalized so that the largest map size used all positions, while smaller mazes used a subset of the map.

**LSTM Agent Details:** The LSTM agent had a single 128-dimension LSTM layer on top of the state embedding. The auxiliary state information was first processed into a 256-dimension embedding.

**MQN Agent Details:** Each past history state was 512-dimensional (256-dimensional key + 256-dimensional value feature). The input state was processed by a convolutional network with 32 channels, filter size 3, stride 1 and padding 1.

## C   3D ENVIRONMENT DETAILS

The state input for all 3D environments was a 100x60 RGB+D image. This was passed through a convolution network that was the same for all architectures. It first consists of a 2D convolution with 32 channels of filter size 8 with stride 4. This was then passed to another 2D convolution with 64 channels of filter size 4 and stride 2. Finally, the result was passed through a fully-connected layer with 512 features, which represented the current frame embedding. The frame embedding was then augmented with some auxiliary information about the map, which in the case of Neural Map, FRMQN and LSTM architectures was 1) a one-hot encoding of the current time step, 2) the current orientation (North/East/West/South), 3) the 2D velocity (change in x/y position in a top-down 2D quantized grid of possible environment positions), and 4) a one-hot encoding of the agent's current quantized position. For Ego Neural Map, only 1, 2, 3 are used (i.e. it has no input of the agent's current true position given by an oracle). The 512 frame encoding is concatenated with the auxiliary state information to form the complete state embedding. Actions in the maze consist of moving forward and turning left or right.

For optimization, all architectures used the Adam optimization algorithm with gradients thresholded to a norm of 40. We used an auxiliary weighted entropy loss on the Synchronous Actor-Critic with weight 0.01. The learning rate for LSTM models was 0.0005, while for other architectures it was set to 0.00075. The hyperparameters were obtained during a limited hyperparameter sweep on a simpler version of the Indicator Maze environment. We trained A2C with a number of steps equal to the episode length (no truncated backprop). We trained the agent for 3000 steps, where each step consisted of a gradient obtained from 100 full episodes. The effective batch size was thus upper bounded by $500 * 100 = 50000$. We used multithreading to calculate the batch gradients efficiently. Each update step took on the order of 1-3 minutes depending on the number of threads available, meaning each agent took on the order of a week to train.

The mazes were generated using an algorithm based on Depth-First Search. Once the completed fully-connected maze was generated, random walls were deleted with a probability $p$. This $p$ probability was chosen at maze creation time and was sampled uniformly from between 0.0 and 0.6. Mazes were between size 4 to size 8. The size of a maze represents how many "cells" there are in the maze, where a cell is an area which can potentially have walls on each side. A set of 50 test maze geometries were sampled to act as a test set, and were made sure to never be sampled during training. These 50 test mazes were generated with $p = 0$, so they represent the most difficult mazes seen during training due to their higher degree of partial-observability. Goal locations were sampled uniformly at random when the mazes are generated.

**Indicator Maze Details:** The indicator mazes used a curriculum approach to accelerate learning. In 50% of the episodes sampled, only one goal existed in the environment and the indicator color matched the single goal color, making entering an incorrect goal impossible. This curriculum prevented the agents from learning to always enter a single color goal, which happened often when learning on only double goal environments. The test environments only used double goals. A reward of +1 was given to correct goal entry, and a negative reward of -1 was given to incorrect goal entry or episode terminating after a maximum number of time steps. These time steps depended on the maze size and were [150, 250, 300, 400, 500] for maze sizes [4, 5, 6, 7, 8]. At test time, the maximum time steps were extended to [300, 500, 600, 800, 1000].

**Repeating Maze Details** We included the same curriculum as the indicator maze. Rewards were the same as the indicator maze. After the first time the agent reaches a goal, the red torch is shown at each subsequent episode (regardless of what the correct indicator was). Maximum time steps were again [150, 250, 300, 400, 500] for maze sizes [4, 5, 6, 7, 8]. At test time, the maximum time steps were extended to [300, 500, 600, 800, 1000].

**Minotaur Maze Details:** For minotaur maze, a reward of +0.5 was given when reaching the randomly located goal and a reward of +1.0 was given when returning to the initial position. Episodes are terminated once the agent completes the return path, otherwise a negative reward of -1 is given if the agent exceeds the maximum number of steps. The maximum time steps were again [150, 250, 300, 400, 500] for maze sizes [4, 5, 6, 7, 8]. At test time, the maximum time steps were extended to [300, 500, 600, 800, 1000].

| Agent |  | Indicator | | | Repeating | | | Minotaur | | |
|---|---|---|---|---|---|---|---|---|---|---|
| Maze Size | | 4 | 5 | 6 | 4 | 5 | 6 | 4 | 5 | 6 |
| $\sigma = 0$ | Acc | 88.4 | 84.4 | 79.3 | - | - | - | 97.3 | 89.0 | 62.0 |
| | Rew | - | - | - | 9.47 | 9.91 | 5.91 | 1.46 | 1.34 | 0.93 |
| $\sigma = 0.01$ | Acc | 92.4 | 91.9 | 84.3 | - | - | - | 96.7 | 86.2 | 66.1 |
| | Rew | - | - | - | 12.4 | 12.9 | 10.9 | 1.45 | 1.29 | 0.99 |

Table 3: Results on the three 3D Doom maze tasks for the fixed-frame Neural Map with Controller LSTM. We can see that adding small compounding error does not largely affect the ability of the Neural Map to learn memory tasks and even has a beneficial effect for some tasks. Hyperparameters and architectures used were the same as presented in the main results.

(Ego-)**Neural Map Agent Details:** For the global read operation, we used a convolutional network with 3 layers of 8 channels and kernel size 3. The strides were set to 1 on the first layer, and 2 to the second and third layers. Padding was 1 on the first layer and 0 on other layers. The 3 convolutional layers were then followed by a fully-connected layer of dimension 256 and then another fully-connected layer of size 32. All activations were relu except the 32 dimension layer, which was set to tanh. The Neural Map itself was size 32x15x15. Positions were normalized so that the largest map size used all 15x15 positions, while smaller mazes used a subset of the map. Additionally, during writing we split the 32 channels of the map into 8 channels per orientation (so if the agent is facing north, it writes only to the first 8 dimensions, if south, the next 8, and so on).

**LSTM Agent Details:** The LSTM agent had a single 256-dimension LSTM layer on top of the state embedding.

**FRMQN Agent Details:** Each past history state was 64-dimensional (32-dimensional key + 32-dimensional value feature). Due to large matrix multiplies from storing the entire episode history, having larger feature sizes causes the attention operation to start becoming prohibitively expensive.

## D   NEURAL MAP WITH DRIFT NOISE MODEL

We did an additional experiment on the Neural Map that featured drift noise to simulate the effects of the agent using a local visual odometry model that had small error in predicting each frame-by-frame transformation. This is meant to represent a more realistic scenario (e.g. robotic navigation) where perfect localization is not feasible but a relatively accurate estimate can be provided, demonstrating the robustness of the architecture to noise. For example, we could assume the Neural Map is run in parallel with a SLAM algorithm which provides an estimate of the agent's current position.

To model this noise, we add a zero-mean gaussian random variable to the oracle position with a variance that depends on the current time-step. In more detail, the noise-corrupted positions $(\hat{x}, \hat{y})$ in an $W \times W$ size map provided to the Neural Map are:

$$(\hat{x}, \hat{y}) = (\max\{\min\{\lfloor x + \epsilon_x \rfloor, W - 1\}, 0\}, \max\{\min\{\lfloor y + \epsilon_y \rfloor, W - 1\}, 0\}),$$
$$\epsilon_x, \epsilon_y \sim \mathcal{N}(0, \sigma^2 t)$$

This simulates the effect of an odometry algorithm which has independent zero-mean gaussian error with equal variance. This error compounds over time causing the variance to grow with the time step. We evaluate the Neural Map with noise $\sigma = 1/100$ on smaller versions of the 3D Doom maze tasks (maze sizes [4, 5, 6]) and compare it to the version with perfect odometry. We train for 1500 steps of 100 episodes each step. Results are shown in Table 3. We can see that adding a small amount of error at each time step does not largely affect the results of the memory and can even benefit it, with some noticeable improvements on Indicator and Repeating tasks. It's possible that the noise acts as a regularizer to speed up learning. For the Minotaur task, since positional information is important because the agent must remember the entire path taken, adding noise causes slight decrease in reward in mazes of size 4 and 5, but otherwise performance is very similar. Therefore this means that the Neural Map is likely to work in the case where a localization oracle is not available and instead only error-prone odometry is.

We also plot some example trajectories to compare the effect of noise. We can see that the noise causes some slight aliasing in the position, which increases as time passes. The positions are quantized to a 15×15 grid.

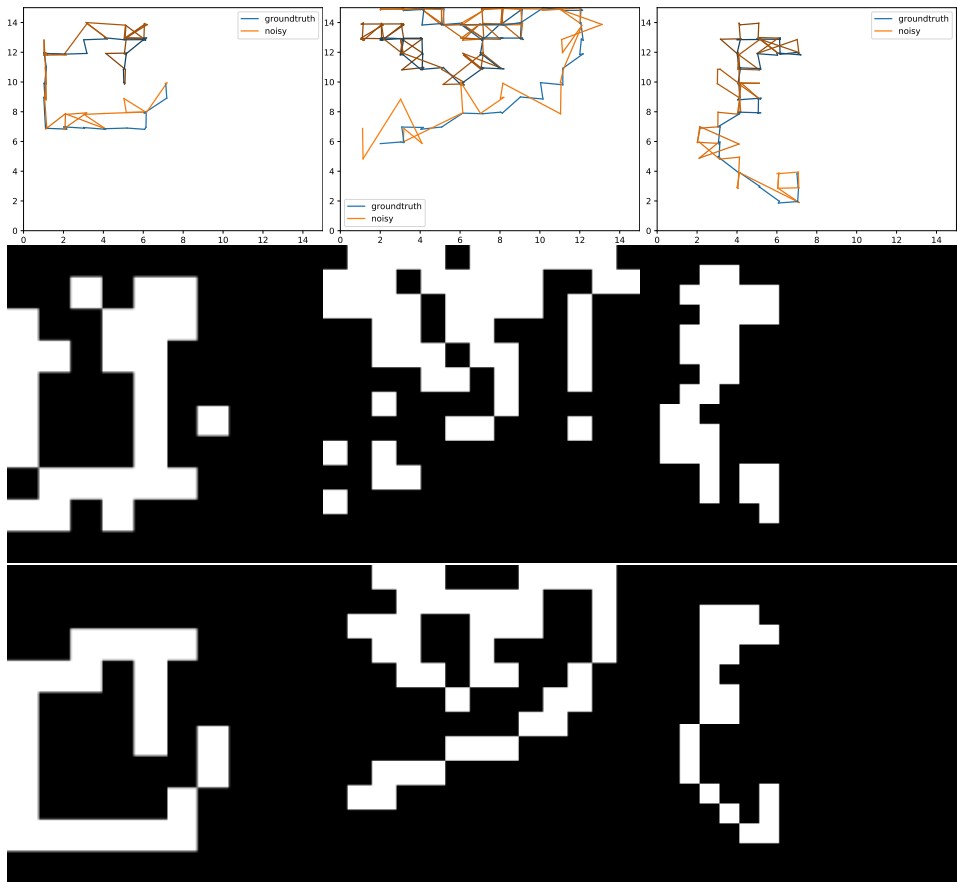

Figure 3: **Top:** Noisy v.s. Groundtruth Position trajectory (quantized to a 15×15 grid). As time progresses, the colors get lighter. **Center:** Neural Map cells addressed by the write operator under the noisy positions. **Bottom:** Neural Map cells that would have been written to under perfect position estimates.

# E  SAMPLES OF CONTEXT READ DISTRIBUTION

## E.1  2D ENVIRONMENT

To provide some insight into what the Neural Map learns, we show samples of the probability distribution given by the context read operation in a 2D maze example. We ran it on an example maze shown in Figure 4. In this figure, the top row of images are the agent observations, the center row are the fully observable mazes and the bottom row are the probability distributions over locations from the context operation, e.g. the $\alpha_t^{(x,y)}$ values defined by Eq. 2. In this maze, the indicator is blue, which indicates that the teal goal should be visited. We can see that once the agent sees the incorrect red goal, the context distribution faintly focuses on the map location where the agent had observed the indicator. On the other hand, when the agent first observes the correct teal goal, the location where the agent observed the indicator lights up brightly. This means that the agent is using its context retrieval operation to keep track of the landmark (the indicator) that it has previously seen.

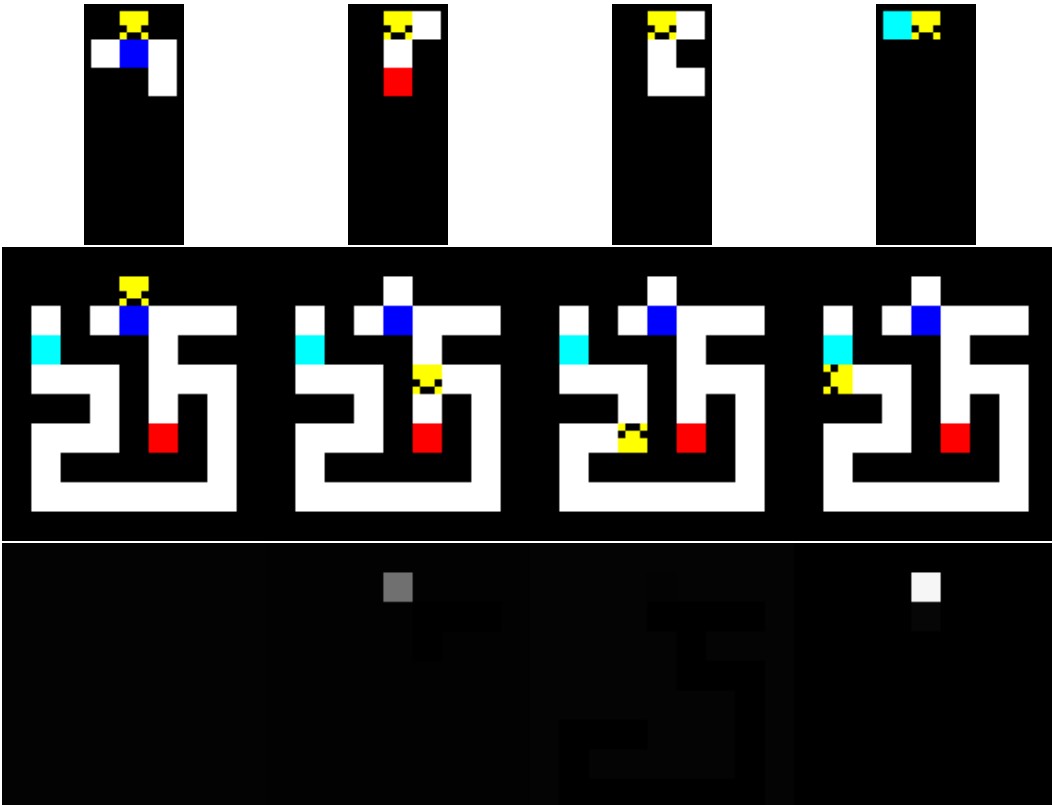

Figure 4: A few sampled states from an example episode demonstrating how the agent learns to use the context addressing operation of the Neural Map. The top row of images is the observations made by the agent, the center is the fully observable mazes and the bottom image is the probability distributions over locations induced by the context operation at that step.

## E.2  3D ENVIRONMENT

We draw some examples of the context addressing probability distribution in the 3D Doom environment in Figure 5 (allocentric) and Figure 6 (egocentric). We can see that the Neural Map learns to use its context addressing operator to retrieve the indicator torch identity, until it sees the correct corresponding tower. Once it sees the correct tower there is a shift in how the agent uses the map and the probability map seems to invert, addressing the parts of the map that were unexplored. This effect is consistent in both allocentric and egocentric variants. This might be because the Neural Map variant used on Doom had an internal LSTM which could enable it to remember the indicator identity for the short amount of time it took to walk up to the goal.

**Indicator Prediction** To determine whether the Neural Map was accurately storing the indicator identity within its memory, we train a logistic regression model on memory vectors sampled over 75 episodes. We then attempt to predict the indicator on a held-out set of 25 episodes by taking the max prediction over all positions of the memory at the end of the episode. We can see that a simple logistic regression is capable of recovering the indicator in 100% of the episodes, showing that the indicator identity can be easily extracted from, e.g., the context operator.

| Agent | Indicator Accuracy |
|---|---|
| Controller NMap | 100% |
| Controller Ego-NMap | 100% |

Table 4: Figure showing accuracy of a logistic model to determine indicator identity from the stored Neural Map features.

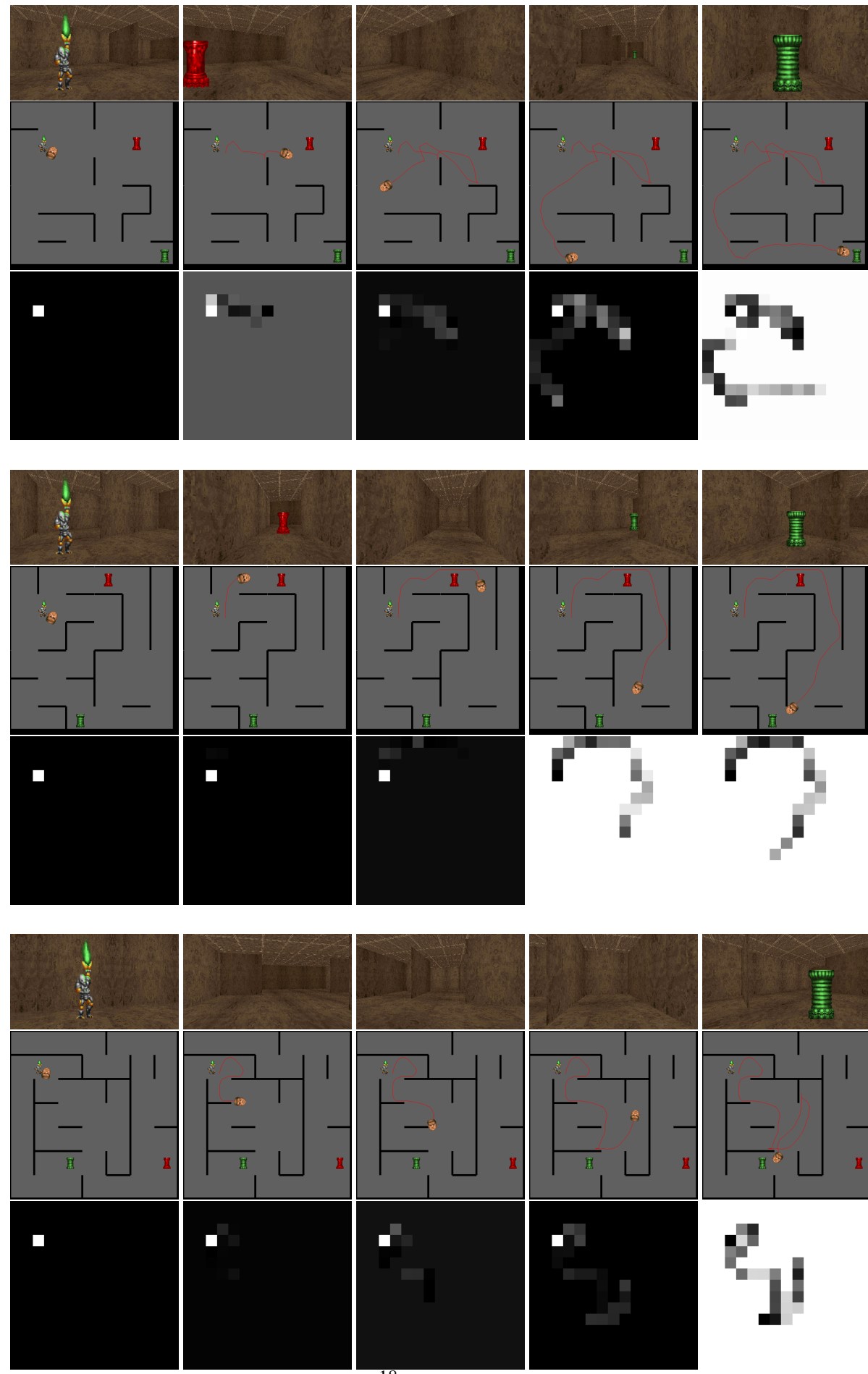

Figure 5: Three example episodes of the (allocentric) context addressing operator on Doom mazes. The top images of each row are the RGB inputs the agent sees, the center images are a top-down representation of the maze, and the bottom images are the $\alpha_t^{(x,y)}$ of the context operation.

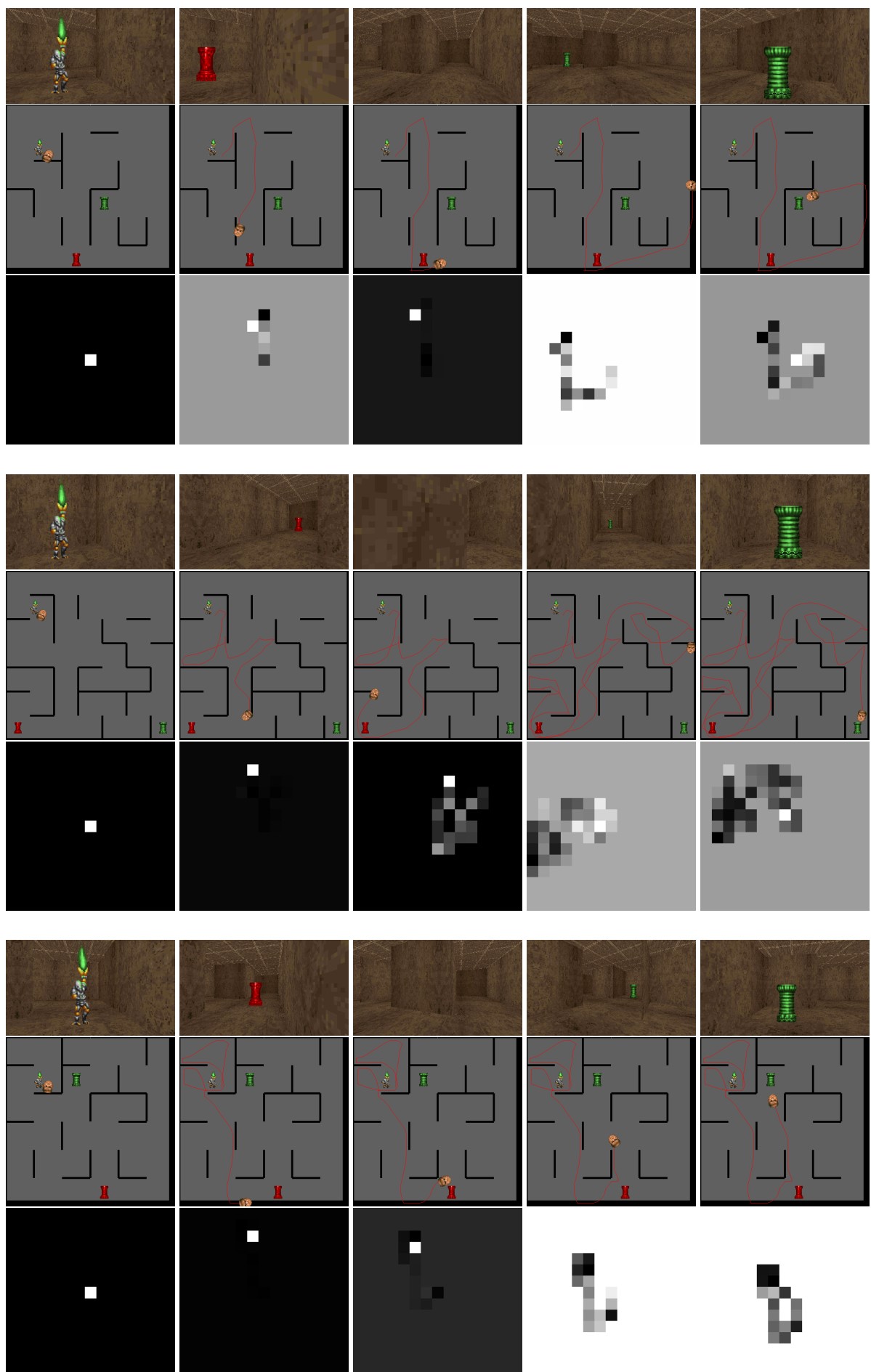

19

**Figure 6:** Three example episodes of the (egocentric) context addressing operator on Doom mazes. The top images of each row are the RGB inputs the agent sees, the center images are a top-down representation of the maze, and the bottom images are the (egocentric) $\alpha_t^{(x,y)}$ of the context operation.

## F  BACKTRACKING

We also explored whether the allocentric and egocentric Neural Maps were capable of using their memories in order to do backtracking, i.e. re-visiting unexplored areas of the maze. To measure this, we developed a variant of the Indicator Maze where the goal states were removed. We want to measure how much of the maze is explored by the agent under this setting where there are no terminal states. To measure how much of the maze was explored, we quantized the 50 test mazes into 11 discrete positions and counted how many of the quantized positions the agent visited. We report results below in Table 5

| Agent | Visitation Score |
|---|---|
| Controller NMap | 71.6% |
| Controller Ego-NMap | 77.6% |
| LSTM | 68.5% |

Table 5: Visitation scores of the Neural Map models which measure how much of a maze is explored within a set time limit. We can see that the egocentric neural map explores more of the mazes than the allocentric model, exploring on average 77.6% of the test mazes. The allocentric neural map explores 71.6% of the test mazes. The LSTM is reported to provide a point of comparison.

