# OpenReview forum: " Neural Map: Structured Memory for Deep Reinforcement Learning"
_ICLR.cc/2018/Conference — Accept (Poster)_

### Official Review · AnonReviewer1 · 2017-11-27
**OK paper--needs further ablation experiments**

**Rating:** 7
**Confidence:** 4

**Review:**

The paper introduces a new memory mechanism specifically tailored for agent navigation in 2D environments. The memory consists of a 2D array and includes trainable read/write mechanisms. The RL agent's policy is a function of the context read, read, and next step write vectors (which are functions of the observation). The effectiveness of the proposed architecture is evaluated via reinforcement learning (% of mazes solved). The evaluation included 1000 test mazes--which sets a good precedent for evaluation in this subfield.

My main concern is the lack of experiments to test whether the agent really learned to localize and plan routes using it's memory architecture. The downsampling experiment in Section 5.1 seems to indicate the contrary: downsampling the memory should lead to position aliasing which seems to indicate that the agent is not using its memory to store the map and its own location. I'm concerned whether the proposed agent is actually employing a navigation strategy, as seems to be suggested, or is simply a good agent architecture for this task (e.g. for optimization reasons). The short experiment in Appendix E seems to try and answer this question, but it's results are anecdotal at best.

If good RL performance on navigation tasks is the ultimate goal then one can imagine an agent that directly copies the raw map observation (world centric) into memory and use something like a value iteration network or shortest path planning to plan routes. My point is that there are classical algorithms to solve navigation even in partially observable 2D grid worlds, why bother with deep RL here?

---

> ### Author Response · Authors · 2017-12-22
> **Reviewer 1 Reply**
>
> Dear Reviewer 1,
>
> We thank you for your valuable comments and feedback. With respect to the concern over the lack of experiments, we have run experiments on 4 different memory-based environments and on each environment shown that the Neural Map exceeds the performance of previous baseline models, including LSTMs and Memory Networks. We think this has sufficiently demonstrated that the Neural Map demonstrates a performance improvement on memory-based navigation tasks.
>
> We have also added additional results in appendix E demonstrating more episodic examples of the context-based retrieval on 3D tasks, including both egocentric and allocentric versions of the Neural Map. From these results, we can see that the Neural Map uses its context operator to mostly retrieve states around the starting position where the indicator is in full view. In addition, we further demonstrated that the indicator identity could be inferred with 100% accuracy from the memory map using just a logistic regression model. To explore whether the Neural Map used its memory to accurately plan routes, we measured its ability to do backtracking. We showed that the egocentric variant of the Neural Map explores on average around 10% more of the test mazes compared to  an LSTM baseline.
>
> With respect to the downsampling experiment in Section 5.1, each wall in the environment takes one 'pixel' in the map, so the reduction to 8x8 is only aliasing on average 2 positions compared to the 15x15 map. We argue that this is not a significant enough reduction in spatial resolution to cause a large decrease in performance, and the Neural Map can still navigate at this slightly larger spatial scale. The fact that, comparatively, the 6x6 map decreases significantly in performance due to larger aliasing (aliasing up to 3x3 positions) provides evidence that the Neural Map does utilize spatial information to navigate, but is robust to some small noise.
>
> With respect to the point about motivating the use of Deep RL, we believe the "Repeating" environment shows the added capability of using memory-based Deep RL over using only traditional navigation algorithms. In this Repeating environment the indicator always changes to red after the first goal entry, meaning an agent that just writes/maps observations from its current position would not be capable of remembering the original indicator color after the first goal entry  (as on being reset to the initial position after the first goal entry, its observation would be overwritten with a potentially incorrect indicator color).
>
> Similar to the repeating environment, we can envision many other applications of Deep RL within dynamic environments, where the environment is continuously changing. For example, an office environment where objects are constantly being moved and misplaced. In such an environment, a navigation system  on a map of past observations might by itself not be sufficient, and a differentiable memory that writes its own features into memory could potentially learn things such as "if object X is not at Y, it is likely to be at Z" in an end-to-end manner without pre-specification by an expert.

---

> ### Comment · AnonReviewer1 · 2018-01-03
> **Revised rating**
>
> Based on the author's rebuttal I have revised the score to a 7.

---

### Official Review · AnonReviewer3 · 2017-11-27
**Important paper about structured memory for navigation**

**Rating:** 9
**Confidence:** 5

**Review:**

This paper presents a fully differentiable neural architecture for mapping and path planning for navigation in previously unseen environments, assuming near perfect* relative localization provided by velocity. The model is more general than the cognitive maps (Gupta et al, 2017) and builds on the NTM/DNC or related architectures (Graves et al, 2014, 2016, Rae et al, 2017) thanks to the 2D spatial structure of the associative memory. Basically, it consists of a 2D-indexed grid of features (the map) M_t that can be summarized at each time point into read vector r_t, and used for extracting a context c_t for the current agent state s_t, compute (thanks to an LSTM/GRU) an updated write vector w_{t+1}^{x,y} at the current position and update the map using that write vector. The position {x,y} is a binned representation of discrete or continuous coordinates. The absolute coordinate map can be replaced by a relative ego-centric map that is shifted (just like in Gupta et al, 2017) as the agent moves.

The experiments are exhaustive and include remembering the goal location with or without cues (similarly to Mirowski et al, 2017, not cited) in simple mazes of size 4x4 up to 8x8 in the 3D Doom environment. The most important aspect is the capability to build a feature map of previously unseen environments.

This paper, showing excellent and important work, has already been published on arXiv 9 months ago and widely cited. It has been improved since, through different sets of experiments and apparently a clearer presentation, but the ideas are the same. I wonder how it is possible that the paper has not been accepted at ICML or NIPS (assuming that it was actually submitted there). What are the motivations of the reviewers who rejected the paper - are they trying to slow down competing research, or are they ignorant, and is the peer review system broken? I quite like the formulation of the NIPS ratings: "if this paper does not get accepted, I am considering boycotting the conference".

* The noise model experiment in Appendix D is commendable, but the noise model is somewhat unrealistic (very small variance, zero mean Gaussian) and assumes only drift in x and y, not along the orientation. While this makes sense in grid world environments or rectilinear mazes, it does not correspond to realistic robotic navigation scenarios with wheel skid, missing measurements, etc... Perhaps showing examples of trajectories with drift added would help convince the reader (there is no space restriction in the appendix).

---

> ### Author Response · Authors · 2017-12-22
> **Reviewer 3 Reply**
>
> Dear Reviewer 3,
>
> Thank you for the strong support and for your comments and feedback. We understand that the noise model is to some extent simplistic compared to those found in robotics applications, but we argue that it does at least demonstrate that the Neural Map is robust to some degree of drift/aliasing in its position estimate. We have added a figure in Appendix D showing example trajectories from the noisy model.
>
> We have also added an analysis of the memory in the appendix where we demonstrate that the context operator is mainly used to address the positions near the starting state, where the indicator color is in full view. We also demonstrate the improved ability of the Neural Map to explore the test mazes, with the egocentric Neural Map variant exploring on average 10% more than an LSTM baseline.

---

> > ### Comment · AnonReviewer3 · 2017-12-26
> > **Thank you for your rebuttal**
> >
> > Having read the other reviews and rebuttals, I am maintaining a rating of 9 (top 15%, strong accept).

---

### Official Review · AnonReviewer2 · 2017-11-27

**Rating:** 6
**Confidence:** 5

**Review:**

# Summary
This paper presents a new external-memory-based neural network (Neural Map) for handling partial observability in reinforcement learning. The proposed memory architecture is spatially-structured so that the agent can read/write from/to specific positions in the memory. The results on several memory-related tasks in 2D and 3D environments show that the proposed method outperforms existing baselines such as LSTM and MQN/FRMQN.

[Pros]
- The overall direction toward more flexible/scalable memory is an important research direction in RL.
- The proposed memory architecture is new.
- The paper is well-written.

[Cons]
- The proposed memory architecture is new but a bit limited to 2D/3D navigation tasks.
- Lack of analysis of the learned memory behavior.

# Novelty and Significance
The proposed idea is novel in general. Though [Gupta et al.] proposed an ego-centric neural memory in the RL context, the proposed memory architecture is still new in that read/write operations are flexible enough for the agent to write any information to the memory, whereas [Gupta et al.] designed the memory specifically for predicting free space. On the other hand, the proposed method is also specific to navigation tasks in 2D or 3D environment, which is hard to apply to more general memory-related tasks in non-spatial environments. But, it is still interesting to see that the ego-centric neural memory works well on challenging tasks in a 3D environment.

# Quality
The experiment does not show any analysis of the learned memory read/write behavior especially for ego-centric neural map and the 3D environment. It is hard to understand how the agent utilizes the external memory without such an analysis.

# Clarity
The paper is overall clear and easy-to-follow except for the following. In the introduction section, the paper claims that "the expert must set M to a value that is larger than the time horizon of the currently considered task" when mentioning the limitation of the previous work. In some sense, however, Neural Map also requires an expert to specify the proper size of the memory based on prior knowledge about the task.

---

> ### Author Response · Authors · 2017-12-22
> **Reviewer 2 Reply**
>
> Dear Reviewer 2,
>
> We thank you for your valuable comments and feedback.
>
> We have added an analysis of the memory in the appendix E where we demonstrate more episodic examples of the context-based retrieval on 3D tasks, including both egocentric and allocentric versions of the Neural Map. From these results, we can see that the Neural Map uses its context operator to mostly retrieve states around the starting position where the indicator is in full view. In addition, we further demonstrated that the indicator identity could be inferred with 100% accuracy from the memory map using just a logistic regression model.
> To explore whether the Neural Map used its memory to accurately plan routes, we measured its ability to do backtracking. We demonstrated the improved ability of the Neural Map to explore the test mazes, with the egocentric Neural Map variant exploring on average 10% more than an LSTM baseline.
>
> As for setting the memory size, we argue that in many cases it can be easier for an agent designer to specify spatial distances than the time horizon of a task. For example, you could have an agent operating within a household where the agent designer only has to set the spatial extent of the map to represent an area at least as large as the house. On the other hand, estimating how long in time it might take an agent to do a task such as object collection would require knowing things such as e.g. how fast the robot navigates the house, how long it takes to grasp the object, etc.
> Although the memory architecture is limited to 2D/3D environments, we argue that those environments encompass a large portion of real world applications of Deep RL. The Neural Map could potentially be generalized to a memory over graphs but we leave this extension to future work.

---

### Public Comment · (anonymous) · 2017-12-06
**A related work highly inspired by your work**

You might be interested to take a look

Neural SLAM: Learning to Explore with External Memory
(https://arxiv.org/pdf/1706.09520.pdf)

We present an approach for agents to learn representations of a global map from sensor data, to aid their exploration in new environments. To achieve this, we embed procedures mimicking that of traditional simultaneous localization and mapping (SLAM) into the soft attention based addressing of external memory architectures, in which the external memory acts as an internal representation of the environment for the agent. This structure encourages the evolution of SLAMlike behaviors inside a completely differentiable deep neural network. We show that this approach can help reinforcement learning agents to successfully explore new environments where long-term memory is essential. We validate our approach in both challenging grid-world environments and preliminary Gazebo experiments. A video of our experiments can be found at: https://goo.gl/G2Vu5y.

---

> ### Author Response · Authors · 2017-12-22
> **A related work highly inspired by your work**
>
> Thank you for highlighting this related paper. We will add it to the related work section in the final version.

---

### Public Comment · (anonymous) · 2017-12-08
**Memory based network for planning**

You might also want to consider taking a look at Memory Augmented Control Networks (https://arxiv.org/pdf/1709.05706).
This paper uses a DNC style memory along with the Value Iteration Networks. The paper demonstrates strong experimental results. Possible that VIN when combined with DNC overcomes limitations of differentiable memory described as motivation for your work ?

---

> ### Author Response · Authors · 2017-12-22
> **Memory based network for planning**
>
> Thank you for bringing to our attention this related work. This paper was a concurrent submission to ICLR and we were unaware of it before. While it does show that DNCs can do navigation in partially observable environments, it seems that the DNC was trained using supervised learning which can help significantly in training stability. We do not know how it would compare to the Neural Map on more general memory tasks, compared to the partially observable navigation experiments reported in that paper. We will add this paper to the related work in the final version.

---

### Decision · Program_Chairs · 2018-01-29
**ICLR 2018 Conference Acceptance Decision**

**Decision:**

Accept (Poster)

**Comment:**

Biological memory systems are grounded in spatial representation and spatial memory, so neural methods for spatial memory are highly interesting. The proposed method is novel, well-designed and the empirical results are good on unseen environments, although the noise model may be too weak. Moreover, it would have been great to evaluate this method on real data rather than in simulation.